# Defocused Images Change Multineuronal Firing Patterns in the Mouse Retina

**DOI:** 10.3390/cells9030530

**Published:** 2020-02-25

**Authors:** Seema Banerjee, Qin Wang, Chung Him So, Feng Pan

**Affiliations:** Centre for Myopia Research, School of Optometry, The Hong Kong Polytechnic University, Kowloon, Hong Kong, China; seema.banerjee@connect.polyu.hk (S.B.); qinnn.wang@connect.polyu.hk (Q.W.); chunghim.so@polyu.edu.hk (C.H.S.)

**Keywords:** ganglion cell, myopia, amacrine cell, gap junction, retina

## Abstract

Myopia is a major public health problem, affecting one third of the population over 12 years old in the United States and more than 80% of people in Hong Kong. Myopia is attributable to elongation of the eyeball in response to defocused images that alter eye growth and refraction. It is known that the retina can sense the focus of an image, but the effects of defocused images on signaling of population of retinal ganglion cells (RGCs) that account either for emmetropization or refractive errors has still to be elucidated. Thorough knowledge of the underlying mechanisms could provide insight to understanding myopia. In this study, we found that focused and defocused images can change both excitatory and inhibitory conductance of ON alpha, OFF alpha and ON–OFF retinal ganglion cells in the mouse retina. The firing patterns of population of RGCs vary under the different powers of defocused images and can be affected by dopamine receptor agonists/antagonists’ application. OFF-delayed RGCs or displaced amacrine cells (dACs) with time latency of more than 0.3 s had synchrony firing with other RGCs and/or dACs. These spatial synchrony firing patterns between OFF-delayed cell and other RGCs/dACs were significantly changed by defocused image, which may relate to edge detection. The results suggested that defocused images induced changes in the multineuronal firing patterns and whole cell conductance in the mouse retina. The multineuronal firing patterns can be affected by dopamine receptors’ agonists and antagonists. Synchronous firing of OFF-delayed cells is possibly related to edge detection, and understanding of this process may reveal a potential therapeutic target for myopia patients.

## 1. Introduction

The World Health Organization (WHO) report estimated that over 50% of the global population will have myopia by 2050 [1]. The potential productivity loss associated with this burden of vision impairment in 2015 was estimated at US$244 billion. A further US$6 billion in losses would accrue from one of the long-term effects of high myopia-macular/retinal degeneration [2].

Despite its major public health impact, the etiology of myopia is poorly understood. Ocular refraction depends primarily on axial length, corneal curvature, lens power and anterior chamber depth [3]. In myopia, the eye is relatively long for the optical power of the cornea and lens, resulting in distant images focusing in front of the photoreceptors. There is ample evidence to suggest that the retinal cells can sense the focus of images and can then generate signals to regulate eye growth during refractive development [4,5]. The induction of form-deprivation myopia by goggle wearing or lid suturing in chick [6,7], tree shrew [8], mouse [9] monkey and marmosets [10,11] had demonstrated visual feedback in eye growth control. Interestingly, both monkeys [10] and chicks [12] developed myopia even after optic nerve sectioning to separate the eye from the brain. Another myopia induction method, the wearing of defocusing spectacle lenses to shift the image plane in front of or behind the retina, also induced compensating changes in eye growth to reposition the retina at the image plane in chicks [13], mouse [9,14], marmosets [15] and monkeys [16]. These studies have provided strong evidence that retina exposure to defocused images results in altered eye growth and refraction [17,18].

It has been suggested that subsets of retinal ganglion cells (RGCs) respond to focused/defocused images and thus may play pivotal roles in modulating growth and refraction and, hence, myopia development in the retina [19]. As studies have demonstrated that the retina detects signs of defocus during the visually guided refractive development period [20], retinal signaling is a strong candidate for mediating the retina to sclera signaling pathway in refractive development [21,22], ultimately leading to myopia. Therefore, a critical step toward revealing mechanism of myopia is to understand how the firing patterns of the relevant population of RGCs are affected during early stages of myopia development, when defocused images are projected. Currently, the exact effects of defocused images on the firing pattern of these population of RGCs remain unknown.

Dopamine released in the retina solely by dopaminergic amacrine cells in the light [23,24,25] plays a key role in visual processing, synaptic formation, synaptic transmission, and light adaptation [26,27]. Dopamine D1 receptors are located on horizontal cells, bipolar cells, amacrine cells (ACs) and RGCs [28,29]; dopamine D2 receptors are expressed by photoreceptors, ACs and RGCs [27,30,31,32]. Dopamine is arguably involved in visual experience-modulated eye growth [33,34]. In addition, dopamine and dopamine D1 receptor were showed to play a key role in myopia development in the mouse retina [34,35]. Therefore, our goal was to examine how the disturbance of dopamine signaling affects the firing pattern of population of RGCs that partake in focus/defocus detection.

Multi-electrode array (MEA) recordings allow for the detection of many RGCs or displaced amacrine cells (dACs) simultaneously and directly to compare their spontaneous as well as light evoked spiking activities in a local retinal region [36]. In this study, MEA recordings were performed to identify major types of RGCs/dACs with distinct responses to light (ON sustained or transient; OFF sustained or transient; ON–OFF; and ON or OFF delayed). We utilized monochromatic organic light-emitting display (OLED) to project focused and defocused images with programmed spatial modulation on the mouse retina. Cells responses in retinas under focused and defocused images projected were analyzed. Then, their responses were mapped and compared under focused and defocused images for agonists and antagonists of dopamine D1 and D2 receptor application. We also determine if the firing pattern of population of RGCs had any changes under the mimicked myopia status. We found that defocused images can change the excitatory and inhibitory conductance of retinal ganglion cells in cellular level and firing patterns in the population of neurons in the mouse retina. Spatial firing patterns varied with different powers of defocused images and was affected by dopamine receptors. Synchronous firing of OFF-delayed cells is possibly related to edge detection.

Together, our findings raise the possibility that firing pattern of a population of RGCs/dACs in the mouse retina was changed by oscillation between focused and defocused images. We conclude that such change in the population of RGC activities may serve as an early step in myopia development in the retina.

## 2. Results

### 2.1. Characterization of Major RGCs/Displaced AC Populations in the Mouse Retina Based on Their Light-Evoked Spiking Activities

The firing RGCs (n = 2162 RGCs from 25 retinas, Figure 1H) was recorded by using a 256 channels MEA system. First, recorded RGCs were classified based on their light-evoked activities to square wave stimuli (525 nm full field; I = 1311 photoisomerizations per rod per second, (Rh*/rod/sec); 1 s stimulation, 5 s interval). ON and OFF RGCs can be easily separated as they increased their spiking frequencies to either light ON or OFF sets, respectively. Both RGC groups were then further subdivided to either sustained (maintained spiking) or transient (brief spike bursts) populations [37,38]. Our classification scheme, based on RGC responses to full-field square wave stimuli, was comparable to mouse RGC subtypes identified previously by using single-cell recording [37,39]. Five major types were identified in our recordings: ON transient (n = 382, Figure 1A), ON sustained (n = 178, Figure 1B), OFF transient (n = 797, Figure 1E), OFF sustained (n = 333, Figure 1F) and ON–OFF (n = 239, Figure 1D). Further, ON-delayed (n = 21, Figure 1C) and OFF-delayed (n = 211, Figure 1G) RGC types were differentiated based on their delayed response kinetics with >0.3 s latencies. Responses of these RGCs were reminiscent of previously described ON- and OFF-delayed RGCs [40]. Following characterization, firing pattern maps of population of RGCs were created and recorded for ON, OFF, ON–OFF and ON/OFF-delayed RGC responses to specify their positions over the MEA in the following experiments. Similar to our previous findings, there were no statistically significant differences between data using extracellular unit recordings and MEA [41], but it has to be noted that MEA recordings may comprise a mixture of RGC and dACs signals. Therefore, all RGC categories might be composed by both RGCs and dACs. However, this does not affect the overall observation of firing patterns of cell populations in the retina.

### 2.2. Populations of RGCs Firing Patterns Are Changed by Defocused Images

To test whether spike responses of the populations of RGCs change their temporal characteristics and how specific RGCs have changed their response properties when focused images are switched to defocused counterparts, we presented retinas with an image sequence. Programmed 5 × 5 image arrays (Figure 2A) with spatial grating 0.5 Cycle /Degree (C/D) and square-wave grating (each image had diameter 0.6 mm (height) × 0.646 mm (width) and light intensity = 7.4 × 10^4^ Rh*/rod/sec) were projected (1 s stimulation, 5 s interval) onto the surface of the in vitro retina preparation for 10 min, and RGCs firing patterns were recorded (Figure 2B); ON, OFF, ON–OFF and ON/OFF-delayed RGC responses are color coded to label their positions over the MEA. Following the first recording session, the image was switched to 0.2 C/D grating (I = 9.1 × 10^4^ Rh*/rod/sec, 10 min) and the firing patterns were rerecorded and remapped (Figure 2C). Compared with 0.5 C/D grating image recording, 0.2 C/D recordings resulted in a higher number of active units (40 in 0.2 C/D vs. 24 in the first 0.5 C/D experiments). Interestingly however, only 12 cells responded to both 0.5 C/D and 0.2 C/D stimuli at the same position (Figure 2E); of these 12 cells, only 2 cells remained the same cell responses (one cell remained OFF response and one cell remained in the ON–OFF response). The other 10 cells changed to other cell responses (2 OFF cells changed to ON response; 3 OFF cells changed to ON–OFF response; 1 ON cell changed to OFF response; and 4 ON–OFF cells changed to OFF response). The rest of the recorded RGCs responded to either 0.5 C/D (12 cells) or the 0.2 C/D stimuli (28 cells) but not at the same position. Then, the 0.5 C/D image sequence was projected back at the second time to test the consistency of RGC responses. This third firing pattern map was rather similar to the former 0.5 C/D image projection (Figure 2D). In this experiment, a total of 30 RGCs responded to the 0.5 C/D stimulation, which had similar cell numbers to the first 0.5 C/D recording (24 cells), and 15 cells (50%) recorded RGC responses in the same position as the first 0.5 C/D image projection (Figure 2F). Of these 15 cells, 8 cells kept the same responses at the same position (6 OFF cells, 1 ON cell and 1 ON–OFF cell); 7 cells changed to other responses (2 OFF cells changed to ON–OFF; 1 ON–OFF cell changed to ON; and 4 ON–OFF cells changed to OFF cell responses). Totally, 447 cells recorded from 23 retina had responses under 0.5 C/D but increased to 659 cells (147.4%) after 0.2 C/D image projection. Of these cells, only 138 cells (20.9%) had responses in their original positions. After 0.5 C/D was projected again, 316 cells (48%) fired at the same position as the first 0.5 C/D projection.

Next, the populations of RGC firing patterns were then mapped over the MEA while images were presented under their focused states as well as with different dioptric powers of optical defocus (+10D/+20D/−10D/−20D; focused images programmed with diameter 1.804 mm; 0.2 C/D, square-wave grating; light intensities varying from 1.5 × 10^5^ Rh*/rod/sec to 1.1 × 10^5^ Rh*/rod/sec with defocus; 1 s stimulation time at 5 s interval for 10 min. Figure 2G–K). Total number of actively responding RGCs were obtained from the recordings were 29 in focus; 32 with −10D (10 at the same position); 29 with −20D (6 at the same position); 24 with +10D (4 at the same position); and 33 with +20D (7 at the same position).

Then RGC firing pattern was recorded separately for the four different cell populations (ON, OFF, ON–OFF, and ON/OFF-delayed cells) and mapped while focused/defocused status were oscillated. In one particular experiment, we found 12 ON-responding cells, while their number changed when a defocused image was projected: 13/8 with ±10D and 18/4 with ±20D. Of these 12 ON-response cells, 6 cells lost responses in the original units with defocused images; 3 ON-response cells kept the same ON response at the same position, only 2 cells response under −10D and 1 cell response under −20D; 2 ON-response cells changed to OFF-response cells with −10/−20D at the same position; and 1 cell changed to the OFF-response cell under +20D. In the same experiment, we observed 9 OFF-responding RGCs in focused status while the number of OFF RGCs changed to 7/2 under ±10D and 11/1 under ±20D with defocused image projection. Of these 9 OFF-response cells, 5 cells lost cell response at the same position with defocused images; 1 OFF cell changed to an ON response cell at the same position under −10D; 2 OFF cells changed to ON–OFF response cells at the same position under +20D, −10D, and +20D; 1 OFF cell remained OFF response at the same position only with −10D and +20D; 1 OFF cell changed to an ON–OFF cell under −10D; and 1 cell remained OFF response under −20D and +10D but change to ON response cell with +20D. We also found that 6 ON–OFF responding RGCs under focused status changed to 1 under ±10D and 3/9 under ±20D. Of these 6 ON–OFF response cells, 3 cells lost response at the same position after defocus; 2 ON–OFF response cells changed to an ON response with −10D/+20D and +10D. One ON–OFF response cell changed to an ON cell with −10D/+20D and an OFF cell with −20D/+10D. However, only a few ON/OFF-delayed RGCs were encountered in focused status (n = 2) and under +10D (n = 3) and +20D (n = 1) of the defocused status. Interestingly, this relatively low number of ON/OFF-delayed cells increased to 13 under −10D and 23 under −20D (Figure 2L). There is statistical significance (different color asterisk, *p* < 0.01) in the numbers of these four RGC populations changes with oscillation between focused and defocused (+10D/+20D/−10D/−20D) images. But there was no difference after ON or OFF RGCs changed from focus to +10D; OFF RGCs change to +20D and ON/OFF delayed cells changed to +10D/+20D. Totally, 274 cells recorded from 10 retinas in focused image, then, the number of cells decreased significantly (*p* < 0.01) to 129/158 in −10D/−20Dand 185/196 in +10D/+20D defocused image. Of these 274 cells, 135 cells (49%) lost cell responses at the same position with defocused images; the rest of the cells had varied cell responses with defocused images at the same position. The MEA result indicated that RGCs changed their response properties under different stimulation paradigms in the limited area. The next step was to compare the population of RGC firing patterns evoked by focused and blurred images in the mouse retina. In this set of experiments, we programmed the stimulus with a 10 × 10 matrix of spatial grating images. The spatial dimensions of individual images were 270 µm in height and 310 µm in width (Figure 3A–C), and the entire matrix was projected onto the mouse retina. An array of images significantly increased the ability to project images onto the retina in the MEA recording. In addition, the image pattern could be used for tracking and mapping the cell responses.

First, each 0.2 cycle/degree spatial frequency, square-wave grating (Figure 3A,C) in a 10 × 10 image array (square shape; I: 8.6 × 10^4^ Rh*/rod/sec; light stimulus time of 1 s, at 5 s interval recorded for 60 min) was projected onto the mouse retina. Then, to mimic the blurred image in the myopic retina, a 0.2 cycle/degree, (circle shape, Gaussian blur) (Figure 3B,C) an image array (even the light intensity decreased to 6.1 × 10^4^ Rh*/rod/sec; the low-light-sensitive RGCs can still be activated; light stimulus time was 1 s, at 5 s interval and recorded for 60 min) was projected. The reason to select 0.2 C/D here in the experiment is that 0.2 C/D induced relatively more cell responses in MEA. Thus, this will avoid the situation with no cell response due to the extreme spatial frequency (Figure 3D). The populations of RGC firing patterns to 0.2 C/D, square-wave grating (Figure 3E) reflected the pattern of image array projected as RGCs in which putative receptive fields (recording from RGC somata are not necessarily located in the middle of RGC receptive fields) were covered by image matrices displaying spiking activity more often than those with receptive fields falling out of the image array. RGC activity patterns matched the image array pattern projected on the retina. The 0.2 C/D blurred matrices induced almost no RGC light-evoked responses (Figure 3F). Compared with clear focused images, the number of firing RGCs decreased dramatically (from 84 firing cell in 0.2 C/D, square-wave grating to 12 cells in 0.2 C/D, Gaussian blur). No RGC firing pattern matched the array of the projected image (Figure 3F)**.**

### 2.3. Dopamine Effects on RGC/AC Firing Pattern Responses to Focused and Defocused Images on MEA Recording

Dopamine acts as an important neurotransmitter in the retina and mediates retinal development, visual signaling, and refractive development. Dopamine and dopamine D1 receptors have been shown to play a key role in myopia development in the mouse retina [34,35]. Here, the effects of dopamine D1 and D2 receptor agonists and antagonists on the firing pattern of the population of RGCs in the mouse retina were tested.

First, a 5 × 5 image array with each image 0.6 mm (height) × 0.646 mm (width), 0.2 cycle/degree (square-wave grating; light intensity 9.1 × 10^4^ Rh*/rod/sec; stimuli time 1 s in 6-s cycles) was projected onto the in vitro mouse retina (Figure 4A). Following control recordings, either agonists or antagonists of dopamine D1 and D2 receptors were applied. Populations of RGCs were mapped on the MEA in all conditions, and the number of responding cells were compared with the control recordings. Application of the D1R antagonist SCH23390 in a concentration of 5 μM increased the number of responding cells by 157% ± 23% (n = 211 in control; 315 in SCH23390), but only 21.5% ± 6.4% of the recorded cells were fired in both control and experimental conditions at the same position (Figure 4B,I). Application of the D2 receptor blocker eticlopride in a concentration of 25 μM increased the number of responded RGCs as well by 63.3% ± 21.4% (n = 186 in control; 298 in eticlopride), of which 38.2% ± 6.8% were fired under both conditions at the same position (Figure 4E,F,I). Thus, the effects of both the D1 and the D2 antagonists were similar: they increased the population of firing RGCs. In contrast, D1R agonist SKF38393 (10 μM) and the D2 receptor agonist quinripole (100 μM) reduced the number of light-evoked RGCs; SKF38393 decreased the number of RGCs to 59.9% ± 7.6% (n = 221 in control, 131 in SKF 38393; of which 15% ± 2.7% of RGCs were fired at the same position (Figure 4C,D,I), while Quinripole decreased the number of RGCs to 34.8% ± 3.9% (n = 261 in control, n = 109 in Quinripole). Similarly, 7.6% ± 2.1% of RGCs fired in the original position (Figure 4G–I).

### 2.4. Focused and Defocused Images Can Change the Excitatory and Inhibitory Conductance of RGCs in the Mouse Retina

ON alpha, OFF alpha and ON–OFF RGCs represent three fundamental RGC response types to encode the visual information in the vertebrate retina. Various excitatory and inhibitory retinal microcircuits are wired in the retina to establish responses to ON or OFF sets of light stimulation. To investigate whether oscillation between focused/defocused images disrupt the balance between inhibition and excitation at the cellular level of RGCs, EPSCs and IPSCs of ON alpha, OFF alpha and ON–OFF RGCs were examined. This was achieved by holding the membrane potential (V_h_) on the levels matching either the excitatory (V_h_ = −68 mV) or the inhibitory reversal potentials (V_h_ = 0 mV). First, we carried out measurements when focused images were projected and then switched to optical defocused image (± 20 dioptric powers) counterparts (light stimuli were 0.002 cycles/degree gratings, light intensity = 5.09 × 10^4^ Rh*/rod/sec) under a 40× water immersion lens by using a custom-made image projection system as described before [19]. However, this data should be interpreted with caution, as defocusing under microscopy can change the light intensities and receptive field of RGCs. The receptive field can be kept consistent by programing the projected image area, and the light intensities can be varied but it may not change the whole cell responses [19].

Kinetics of both the EPSC and IPSC currents of ON alpha, OFF alpha and ON–OFF RGCs changed with defocused image projection. In one set of experiments, focused images evoked 142.6 ± 8.9 pA excitatory and 375.2 ± 14.5 pA inhibitory currents from ON alpha RGCs. Defocused images (± 20D) decreased both the excitatory (73.4 ± 1.9 pA under −20D; 49.9 ± 5 pA under +20D) and the inhibitory currents (327.3 ± 3.53 pA under −20D; 120.1 ± 3.2 under +20D) of 55% of ON alpha RGCs (6 of 11) (Figure 5A). This decrease of EPSC/IPSC was statistically significant induced by defocused image projection on ON alpha RGC population (*p* < 0.01) (Figure 5D).

Most OFF alpha RGCs (7 of 9, 78%) also decreased in IPSCs (269.3 ± 8.3 pA under −20D; *p* < 0.05; 308.2 ± 8.5 pA under +20D; *p* < 0.01) under defocused images compared to focused image counterparts (346.7 ± 6.8 pA) (Figure 5B). However, there was no significant difference (*p* > 0.05) in EPSCs when defocused images (197 ± 15 pA under −20D; 184.5 ± 16.5 pA under +20D) projected were compared with focused images (235 ± 3.8 pA) (Figure 5D).

Defocused images also changed response profiles of ON–OFF RGCs (10 of 13, 77%) but displayed a disparity between the ON and OFF response components. The ON response component showed decrease in EPSCs (11.7 ± 8.9 pA; *p* < 0.05) under +20D defocused image stimulation but had no change (37.6 ± 4 pA; *p* > 0.05) under −20D compared to the focused status (43.7 ± 3 pA) (Figure 5C). In contrast, IPSCs of the ON response decreased to 25 ± 3 pA (*p* < 0.05) under the −20D defocused condition compared with the focused status (61.3 ± 6.4 pA), whereas no change was observed (49.7 ± 0.9 pA; *p* > 0.05) when +20D defocused images were applied. The OFF-response components of ON–OFF RGCs showed a significant decrease in EPSCs (37.9 ± 1.1 pA under −20D; 30 ± 4.5 pA under +20D; *p* < 0.01) under defocused images when compared to those of focused images (64.3 ± 2.3 pA). Contrarily, there was no significant change of IPSCs between defocused (90 ± 5.2 pA under −20D; 99.3 ± 4.3 pA under +20D) and focused images projected (97.1 ± 1.6 pA) (Figure 5D). Thus, oscillation between focused/defocused images affected ON and OFF components of ON–OFF RGCs in a different manner.

### 2.5. OFF-Delaying RGCs Synchronized Firing May Contribute to Image Edge Detection

It has been shown that the retina can rapidly and reliably transmit spatial information encoded by spike trains of RGC population to the brain [42]. In the previous experiments, ON- and OFF-delayed RGCs displayed increased spiking activities when an image defocused (Figure 2K). To examine the potential role of ON/OFF-delayed RGC population in myopia, OFF-delayed RGC spikes were recorded under 525-nm full-field stimulation (light intensity 1311 Rh*/rod/sec; stimulus time 1 s, at 5 s interval —the same stimulation was used in MEA recordings). We identified OFF-delayed RGCs based on more than 0.3-s latency light responses as was determined in the previous experiment (Figure 6A,B). Then, the cell was visualized by Neurobiotin injection and double labelled with anti-ChAT antibody to identify the stratification (Figure 6C).

To determine the synchronized firing activity of cell pairs, cross-correlogram profiles (CCPs) for the light-evoked responses were generated, which revealed correlated activity as histogram peaks exceeding chance above the 99% confidence level. To demonstrate spike correlations between OFF-delayed RGC pairs that were not time-locked to the light stimulus, data were time shuffled using a shift-predictor protocol, which was then subtracted from the original CCP.

Two synchronized firing patterns were observed (Figure 7A,B) between the OFF-delayed RGCs. These two synchronized patterns are dual peak and single peak, which represent the two types of coupled cells between RGC-RGC (Figure 7A, dual peak) or between RGC-dAC or AC-AC (Figure 7B, single peak). Then, one of the OFF-delayed RGCs was used as a reference cell to map the synchronized spatial firing pattern under the focused image (Figure 7C) and with the equivalent of ± 20D defocused image (Figure 7D,E). Only the ON/OFF-delayed RGC/ACs were plotted. The map showed the edge of the image spot projected on the mouse retina. Thus, the focused image (with image diameter 1.804 mm; spatial frequency 0 cycle/degree; light intensity 1.6 × 10^5^ Rh*/rod/sec) was roughly reflected in the map of the mouse retina. The size of the MEA array used here was 100/30 µm.

With −20D defocus, the number of ON/OFF-delayed RGCs/ACs decreased dramatically. With the same reference cell, the map between the synchronized firing cells showed that the area plotted also dramatically decreased (Figure 7D). When the +5D defocused image was projected, no image area could be identified due to insufficient cell responses (Figure 7E).

## 3. Discussion

Myopia (near sightedness) is a significant public health problem, affecting over 80% of adults in Hong Kong and 22% of the global population [43]. Emmetropization is a term to describe the active process, in which the expanding eye adjusts to match the powers of the cornea and lens during the early postnatal stages of eye development. Any failure of emmetropization results in refractive errors [44].

Genetics and the environment have been reported to contribute to the development and progression of myopia. Defocusing influences the transcription factor ZENK and thus causes myopia through eye growth [45]. Circadian rhythms and outdoor activities also play a role in myopia development. Although, the etiology of myopia development is still not entirely clear, it is well established that image blur or defocused images alter eye growth and refraction and that these processes are largely governed by the retina [46,47]. It has also been shown that myopia can be induced even after the eye was disconnected from the central nerve system (CNS) [12]. It has been proposed that retinal signals mediated the eye’s refractive development via a retina-to-sclera signaling pathway [47,48]. This study in combination with our previous research [19] showed that defocused images change mouse RGC signaling which might serve as the primary initiators of this retina-to-sclera signaling mechanism.

A precise control mechanism is needed to achieve maximum visual acuity during emmetropization [13]. Retina signaling could be the first and the key role. ON and OFF responses are the most significant visual features encoded by RGCs in parallel information processing in the retina [49,50]. Compared with focused images, defocused images will change the size, focused plane and light intensity. Even though these changes are tiny, the changes will be important for retina to discern the defocused image [19]. The question is whether the RGCs can sense it and code the information into spikes and then send it to the brain. In the current study, ON alpha, OFF alpha and ON–OFF RGC but not all of them showed the changes in excitatory and inhibitory conductance under focused and defocused images. Therefore, biophysical properties of a single cell can sense signals of the presence of defocused image. This result is consistent with our former research that showed defocused images changing the signaling of some ON and OFF alpha RGCs and ON–OFF RGCs in the mouse retina [19].

Interestingly, plus and minus defocused images had different impacts on the current of excitation/inhibition of ON–OFF RGCs. Compared with OFF alpha GCs and ON alpha GCs, ON–OFF RGCs showed diverse responses to plus and minus defocused images. ON–OFF RGCs enable more efficient neural coding, providing better coding for extracting information [50]. Bistratified ON–OFF RGCs have the advantage to be able to detect and code the focused plane over ON/OFF response splitting GCs. In this experiment, ON–OFF RGCs had different response cells to varied defocused images (Figure 2L). ON–OFF RGCs may thus compute and compare information of focused/defocused images.

At the population-cell level, the retina could also reflect the image projected on the retina, leading to a dramatic change in the firing pattern under the defocused image. When the 0.5 C/D image was projected on the retina and then switched back after a different spatial frequency image had been projected, the retina displayed a similar firing pattern as before. The result showed that retina can discern the different spatial frequency images and that RGCs change their response properties under different stimulation paradigms. Notably, the same spatial frequency image did not have the exact firing pattern as before, but it retained a similar pattern in all the trails. It is possible the retina had some complicated mechanism to code and recognize the similar image. The MEA electrode could pick up the same cell signal from soma or dendrite in the receptive field, yet it showed in another electrode of MEA. However, MEA recording of the chicken retina showed that RGCs did not respond differently to defocused images [51]. This might be an animal species difference.

Dopamine acts as an important neurotransmitter in the retina and mediates retinal development, visual signaling and refractive development. Dopamine and dopamine D1 receptor play key roles in myopia development in the mouse retina [34,35]. Dopamine D1 receptor agonist and antagonist applications induced changes in trace coupling of AII-AII ACs, ACs-RGCs and RGCs-RGCs. In this study, D1 and D2 receptor antagonists were found to increase the numbers of firing RGCs whereas agonists decreased it, showing that dopamine through both receptor types control RGC excitability to initiate stimulus-evoked spiking. Because dopamine and its receptors are closely related to myopia development, their effects need to be investigated further as they could be used as a translational method to control retinal signaling in the myopic retina. How dopamine changed the signal of RGCs in myopic retina is an interesting question. We hypothesize that dopamine had an effect on RGCs in focused and defocused status via AII amacrine cell coupling (unpublished data). Thus, ACs will also play an important role in the defocused status.

It has been shown that single spikes can code substantial information about visual stimuli with remarkable temporal precision [52,53]. The retina can rapidly and reliably code spatial information by neural population with relative spike latencies. Two synchronized firing patterns were observed between the OFF-delayed RGCs (dual peak but less than 400-µm distance) [36] and OFF-delayed RGC-coupled ACs (single peak). These ACs might be polyaxonal ACs or wide-field ACs that cover long distances [54,55]. OFF-delayed RGCs may synchronize with other delayed response RGCs/dACs to define the edge of the image area. However, OFF-delayed RGCs lost their synchronized firing under defocused image. Thus, the neural population spikes might reflect the defocused image as a result of losing synchronized firing of the ON/OFF-delayed RGCs/dACs.

In summary, this study showed that the population of RGCs/dACs in the retina can respond differently to focused and defocused images at the single-cell level and that this mechanism might be the substrate for the proposed retina-to-sclera signaling pathway. Therefore, retinal signaling might be the first and the most important step to trigger myopia and may also serve as a continuous key signal in myopia development.

## 4. Methods

### 4.1. Ethical Approval

All animal procedures were approved by the Animal Subjects Ethics Sub-Committee (17-18/65-SO-R-OTHERS, approved on April 04 2018) of the Hong Kong Polytechnic University and complied with the Guide for the Care and Use of Laboratory Animals published by the National Institutes of Health (8th edition).

### 4.2. Animals

Adult mice (postnatal day 28–56) C57BL/6J(RRID: IMSR_JAX:000664) wild-type (WT), n = 78 of either sex, were used in the study.

### 4.3. Retina Preparation

All experiments were performed during daylight hours. The mice were anaesthetized deeply with an intraperitoneal injection of ketamine (Vedno, St. Joseph, MO, USA) and xylazine (Akorn, Decatur, IL, USA) (80 and 10 mg /kg body weight, respectively), and lidocaine hydrochloride (20 mg ml^−1^, Sigma-Aldrich, St. Louis, MO, USA) was applied locally to the eyelids and surrounding tissue. Eyes were removed under dim red illumination and hemisected anteriorly to the ora serrata. The anterior optical structures and the vitreous humor were removed, and the resultant retina–eyecup with sclera attached for MEA recording was placed in a super-fusion chamber with the RGC layer facing down. The image stimulation was projected under the hole of the MEA chamber. Thus, the image was projected through the RGC layer in the retina just like light entering in the eye. For patch recordings, isolated retinas were dissected into four equal quadrants and attached to a modified translucent Millicell filter ring (Millipore, Bedford, MA, USA). The flattened retinas were superfused with oxygenated mammalian Ringer solution, pH 7.4, at 32 °C [56]. The anaesthetized animals were killed by cervical dislocation immediately after enucleation.

### 4.4. Electrical Recording

Extracellular recordings were obtained from neurons using a 256 channel MEA system (Multichannel Systems Gmbh, Germany) that allowed for simultaneously recording from up to 252 retinal cells. In an electrode grid of 16 × 16, 256 electrodes of MEA were used with electrode spacing 200/100 µm and electrode diameter 30 µm (256MEA200/30iR-ITO or 256MEA100/30iR-ITO used in Figure 7 only). The retina was covered by 27.76–12.11 mm^2^. The temperature of the bath in MEA was maintained at 31–33 °C by heating the bottom of the recording chamber and the incoming solution. Retinas were placed on the array for at least 30 min before recording because the amplitude of the recorded spikes usually improved during this period.

All recorded data were stored for offline analyses. Spike trains were recorded digitally at a sampling rate of 20 kHz with MC Rack (Multichannel Systems Gmbh, BW, Germany). For additional offline analysis, Off-line Sorter (Plexon, Dallas, TX, USA) and Neuroexplorer (Nex Technologies, Littleton, MA, USA) software were used.

Spikes were sorted and time stamped from digitized recordings using principal-component analysis. Peri-stimulus time histograms (PSTHs) (5-ms bins) and cross-correlogram profiles (CCPs) (1-ms bins) were generated from the time-stamped spike recordings of RGC pairs using Neuroexplorer software. The significance of correlated spikes above chance was determined as those correlations exceeding the 99% confidence intervals. To correct for spike correlations between cell pairs that were time locked to the stimulus presentation, the spike data were time shuffled using a shift predictor analysis, which was then subtracted from the original CCP to create a shift predictor CCP. The shift predictor CCP thus provided spike correlations that were temporally independent of the light stimulus. To determine the percentage of correlated spikes between RGC pairs, area under the curve measures were computed for profiles within the shift predictor CCP that exceeded the 99% confidence interval as a percentage of the entire profile in a ±50-ms epoch (Origin; OriginLab Corporation). The number of light-evoked ON and OFF spikes of RGCs or current amplitudes was calculated by a subtraction of the background spike or current activity from those evoked by the light stimulus onset and offset, respectively. Cells were classified as sustained or transient based on spike frequency parameters as described previously [39,41].

Extracellular recordings were obtained from single retinal ganglion cells in the mid-peripheral retina in the nasotemporal plane. Recordings were performed by using an Axopatch 700B amplifier connected to a Digidata 1550B interface and pCLAMP 10 software (Molecular Devices, Silicon Valley, CA, USA). Cells were visualized with near infrared light (>775 nm) at ×40 magnification with a Nuvicon tube camera (Dage-MTI, Michigan City, IN, USA) and differential interference optics on a fixed-stage microscope (Eclipse FN1; Nikon, Tokyo, Japan). Retinas were superfused at a rate of 1–1.5 mL min^−1^ with Ringers solution, composed of (mM) 120 NaCl, 2.5 KCl, 25 NaHCO_3_, 0.8 Na_2_HPO_4_, 0.1 NaH_2_PO_4_, 1 MgCl_2_, 2 CaCl_2_ and 5 D-glucose. The bath solution was continuously bubbled with 95% O_2_ with 5% CO_2_ at 32 °C. Electrodes were pulled to 5−7 MΩ resistance, with internal solutions consisting of (mM) 120 potassium gluconate, 12 KCl, 1 MgCl_2_, 5 EGTA, 0.5 CaCl_2_ and 10 HEPES (pH adjusted to 7.4 with KOH). This internal solution was used in experiments in which spiking was not blocked. In whole-cell or perforated-patch (electrodes were backfilled with 25 μm β-escin) recordings, to improve the space clamp and to block spiking, an internal solution contained QX-314 (0.5 mM) and caesium methanosulfonate instead of potassium gluconate. Absolute voltage values were corrected for 11-mV liquid junction potential in the caesium-based intracellular solution. The excitatory and inhibitory current responses were recorded approximately at the chloride or cation equilibrium/reversal potentials −68 and 0 mV, respectively [41].

As to the concern of outer retinal signaling involvement, all perforated patch-clamp experiments will be performed in the presence of the glutamate receptor blockers L-AP4(20 µM) and CNQX (50 µM) in order to reduce noise originating from upstream pathways [57]. Spike trains were recorded digitally at a sampling rate of 10 kHz with Axoscope software and were sorted by using Off-line Sorter (Plexon, Dallas, TX, USA) and NeuroExplorer (Nex Technologies, Littleton, MA, USA) software.

Pharmacology Reagents included SKF38393 and Quinripole from Tocris (Bristol, UK); SCH 23390 and eticlopride obtained from Sigma-Aldrich (St Louis, MO, USA).

### 4.5. Injection of Neurobiotin

The cells were visualized at ×40 magnification, as described above, and were impaled under visual control using pipette tips filled with 4% Neurobiotin (Vector Laboratories, Burlingame, CA, USA) and 0.5% Lucifer Yellow-CH (Molecular Probes, Eugene, OR) in double-distilled water, then back filled with 3 M LiCl. The electrode resistance was ~100 MΩ. The impaled cells were then injected with a biphasic current (+1.0 nA, 3Hz) for 1 min.

### 4.6. Patterned Light Stimulation

A green, organic light-emitting display (OLEDXL, Olightek, China; 800 × 600-pixel resolution, 85 Hz refresh rate) was controlled by an Intel Core Duo computer with a Windows 7 operating system. In this setup, using a Nikon 40× water-immersion objective (CFI Apo 40× W NIR, NA = 0.8), the area of the retina that received light stimuli was 250 µm in diameter. Under the 40× objective, the 15-µm diameter pixels of the OLED projected and presented 0.25 µm/pixel on the retina for patch recording. Spatial frequency stimuli were generated by PsychoPy onto the photoreceptor layer. The background light intensity was 700 isomerizations Rh*/rod/s, and the highest stimulus was 1.816 × 10^5^ Rh*/rod/s. At this level of background illumination, the rod pathway has been shown to be saturated, leaving the cone pathway to mediate the light response [58]. The system projected defocused images in front or behind the outer segments of photoreceptors to mimic plus and minus defocusing by moving up and down a 40× immersion lens of microscopy. The details of the system and light-projected pathway have been published previously [19].

For the MEA experiment, images emitted from OLED will illuminate directly on the electrode layer of an MEA chamber through the diameter of an 8-mm hole via a custom-made Badal system. An OLED was mounted on a micrometer to move on the rail with plano-convex lenses via a prism to project focused and defocused images below the electrodes of the MEA chamber. To ensure that the retina was stimulated by images projected on OLED, the optical axis of the projection lens, prism and lens on the optical rail was carefully checked before the experiment and not touched during experiment.

### 4.7. Immunocytochemistry

Antibodies: Goat anti-ChAT (1:500, Millipore, St. Louis, MO, USA; Cat# AB144P, RRID: AB_2079751) was used for mouse retina.

For the mouse retina, retinas were obtained from the dorsal section of the mid-peripheral retina in the nasotemporal plane. The retinal pieces attached with filter paper (RGCs up), both after injection and isolated from the eyecups, were submersion-fixed in 4% paraformaldehyde in 0.1M PB, pH 7.5 for 30 min at room temperature. After fixation, the retinas were separated from the filter paper and washed with PBS. After fixation, the tissues were washed extensively with 0.1 M phosphate buffer (PB, pH 7.4) and blocked with 3% donkey serum in 0.1 M PB with 0.5% Triton-X 100 and 0.1% NaN_3_ overnight. The antibodies were diluted in 0.1 M PB with 0.5% Triton-X 100 and 0.1% NaN3, containing 1% donkey serum. The tissues were incubated in primary antibodies for 3–7 days at 4 °C and, after extensive washing, incubated in secondary antibodies overnight at 4 °C. After washing with 0.1 M PB, the tissues were mounted in Vectashield (Vector Laboratories, Burlingame, CA, USA) for observation.

### 4.8. Imaging and Data Quantification

Retinal whole mounts were acquired on a ZEISS LSM 800 with Airyscan (Zeiss, Thornwood, NY, USA) confocal microscope using a 40× objective (N.A. 0.8).

Statistical analyses were performed by using Origin software (OriginLab, Northampton, MA, USA). All data are reported as means ± S.E.M. Statistical significance (*p* < 0.05) was determined by using Wilcoxon–Mann–Whitney rank sum test unless otherwise specified.

## Figures and Tables

**Figure 1 cells-09-00530-f001:**
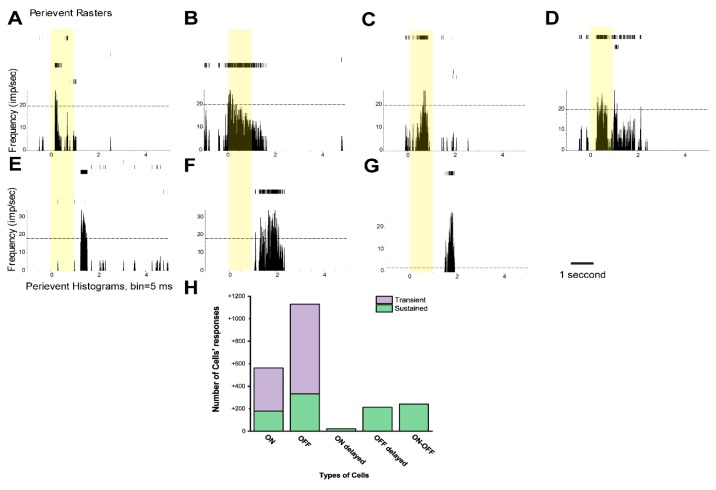
Major types of cells’ responses were identified based on their response profiles to light in the mouse retina. Raster plots and peristimulus time histograms (PSTHs) of cells’ responses were recorded with a multi-electrode array (MEA): (**A**) ON-transient retinal ganglion cell (RGC) to a 1-s full-field light stimulus with its Raster plots (upper part) and PSTH (lower part), (**B**) ON-sustained RGC, (**C**) ON-delayed RGC, (**D**) ON–OFF RGC, (**E**,**F**) OFF-transient RGC and OFF-sustained RGC and (**G**) OFF-delayed RGC or displaced amacrine cell (AC). A 525-nm full-field (light intensity 1311 Rh*/rod/sec) 1-s light stimulus was applied. (**H**) Summary of numbers inside histograms represent recorded cells.

**Figure 2 cells-09-00530-f002:**
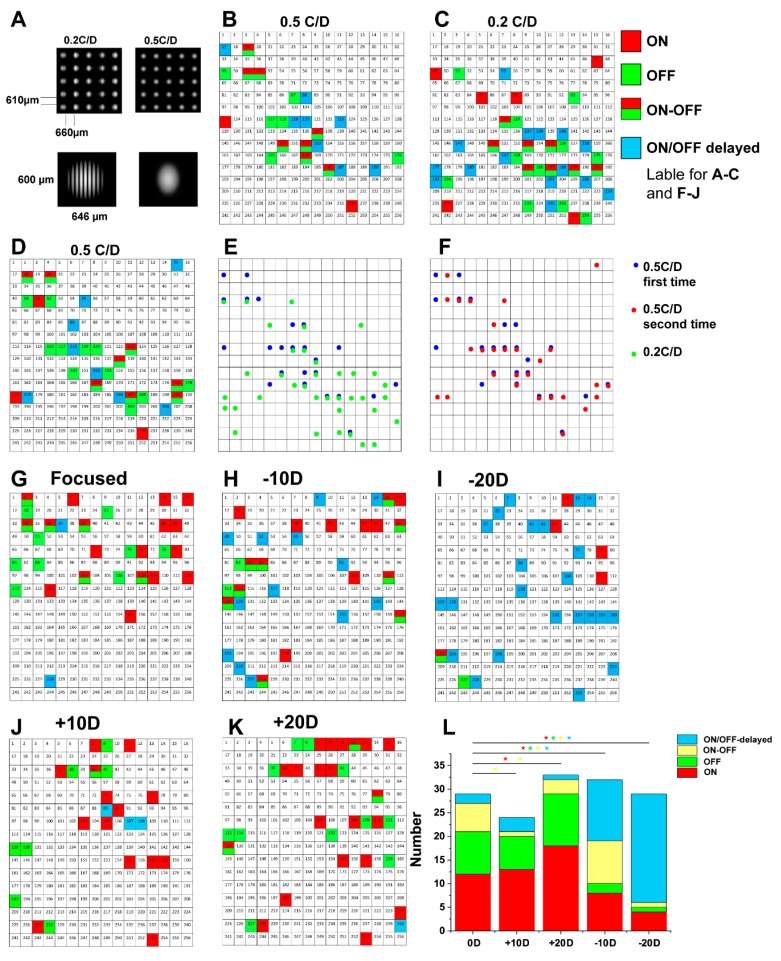
RGC firing responses to different defocused images on MEA recording: Firing pattern recorded from MEA can reflect the image projected on the mouse retina. First, 5 × 5 images (diameter 0.6 mm (height) × 0.646 mm (width), 0.5 cycle/degree (light intensity 7.4 × 10^4^ Rh*/rod/sec); stimuli time 1 s in 6-s circle for 10 min; there were more stripes compared with 0.2 C/D) were projected (**A**). Different colors labeled the ON, OFF, ON–OFF and ON/OFF-delayed cell responses at the position of MEA arrays and were mapped (**B**). The map of the firing pattern changed after image stimulation switch to 0.2 C/D (light intensity 9.1 × 10^4^ Rh*/rod/sec) (**C**). Merging these two maps showed the co-localization of cells’ responses (**E).** Only 30% of cells had responses at the same position. Then, the same 0.5 C/D image stimulation was projected again onto the retina (**D**). For the same position of the first 0.5 C/D projection, 50% of cells had responses. The firing pattern was similar to the former 0.5 C/D image projection (**F**). From Figure 2G–K, the images were programmed with diameter of 1.804 mm; 0.2 C/D, square-wave grating; light intensities varied from 1.5 × 10^5^ Rh*/rod/sec to 1.1 × 10^5^ Rh*/rod/sec under defocus. Firing patterns of RGCs or dACs in the mouse retina changed among the focused images (**G**) and different dioptric powers of optical defocus (+10D/+20D/−10D/−20D) (**H–K).** (**L**) Summary of numbers representing the recording from four different types of cell responses (ON, OFF, ON–OFF and ON/OFF-delayed cells). The bar graph shows the difference in the number of these four types of cell responses under focus and optical defocus (+10D/+20D/−10D/−20D). Different color asterisks represent the statistical significance (*p* < 0.01).

**Figure 3 cells-09-00530-f003:**
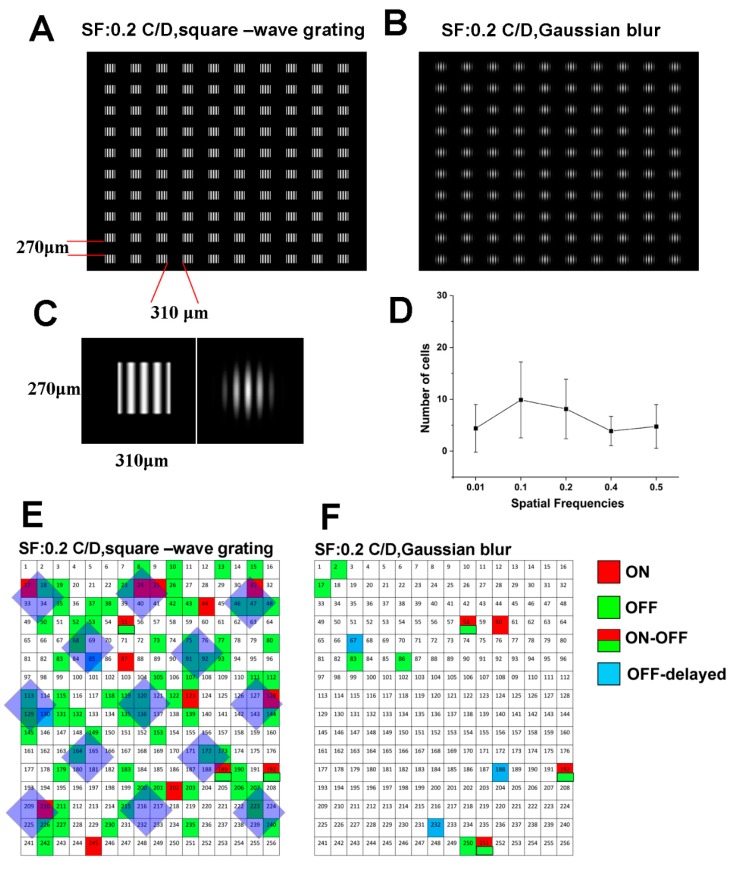
Maps of firing patterns after clear and blurred images were projected on the mouse retina: 10 × 10 image arrays with 270 µm × 310 µm spaftial frequency 0.2 cycle/degree (C/D) (**A**,**C**) clear image, square-wave grating and 0.2 C/D (**B**,**C**) blurred image, Gaussian blur were programmed to project on the mouse retina. (**D**) Cells had maximal responses to 0.2 cycle/degree, square-wave grating with MEA recording. (**E**) Firing patterns of different RGCs/ACs can reflect the image projected for clear images (focused). (**F**) Projection of blurred images could not reflect in the mouse retina. Light intensity: 8.6 × 10^4^ Rh*/rod/sec in 0.2 C/D, square-wave grating to 6.1 × 10^4^ Rh*/rod/sec in 0.2 C/D, Gaussian blur. Light stimulus time was 1 s, at 5 s interval and recorded for 60 min.

**Figure 4 cells-09-00530-f004:**
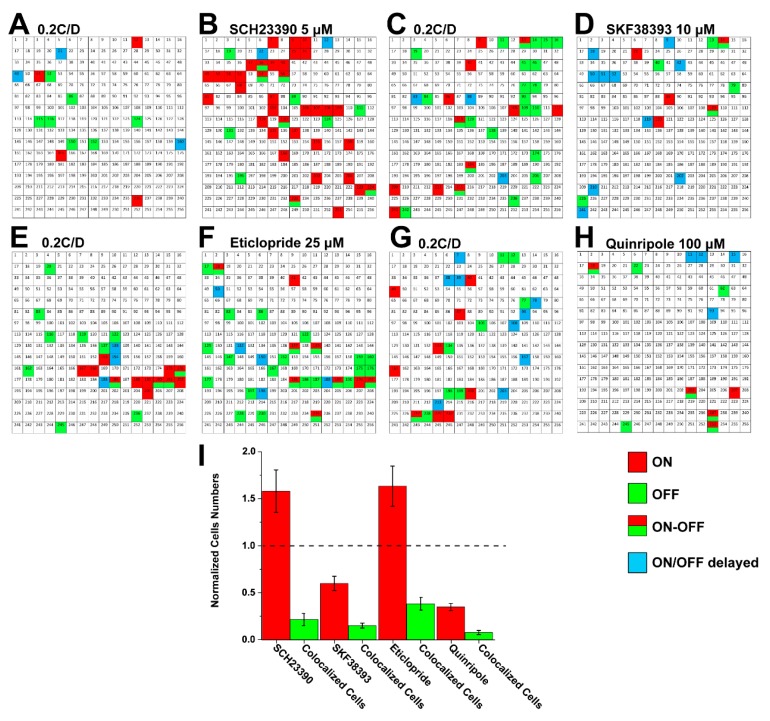
Map of firing pattern changes under dopamine receptors 1 and 2 agonist and antagonist applications in the mouse retina: D1R antagonism SCH23390 5 μM (**A**,**B**) increased firing cell numbers. The same occurred for the D2 receptor blocker eticlopride (25 μM) (**E**,**F**). In the opposite, D1R agonist SKF38393 (10 μM) (**C**,**D**) and D2 receptor agonist Quinripole (100 μM) (**G**,**H**) decreased the firing cells number. Figure 4I summarizes the normalized firing cells number after different agonists and antagonists of dopamine receptor 1 and 2 application. (The dash line is 1 in normalized cells number.) For 5 × 5 image arrays, each image was programmed with diameter 0.6 mm (height) × 0.646 mm (width), 0.2 cycle/degree (square-wave grating); light intensity 9.1 × 10^4^ Rh*/rod/sec; and stimuli time 1 s in 6-s cycles.

**Figure 5 cells-09-00530-f005:**
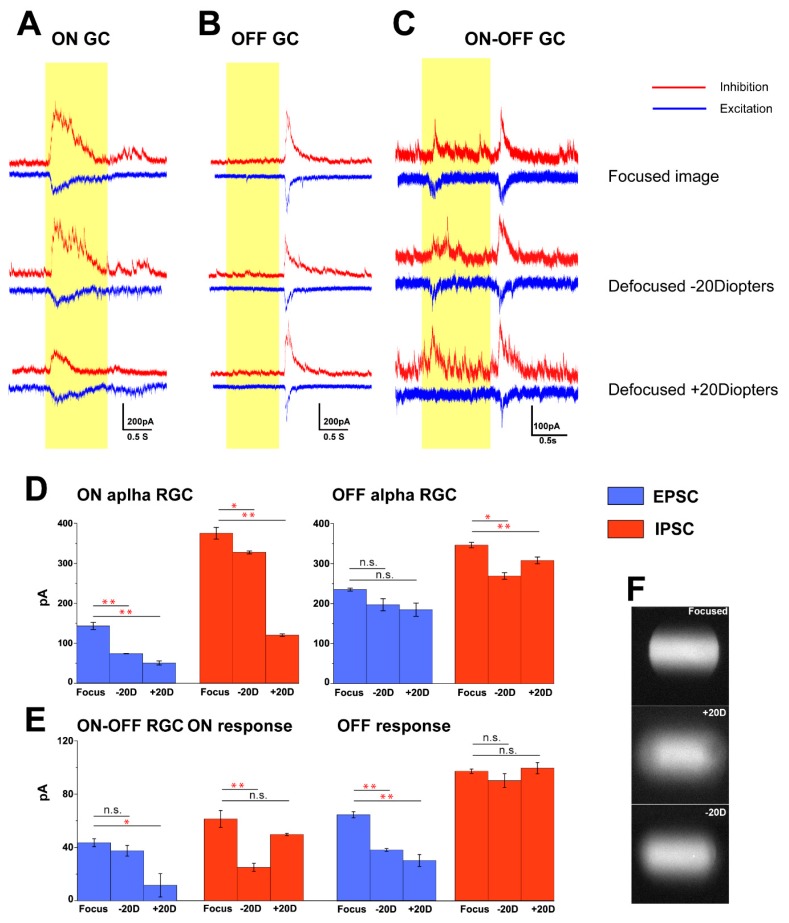
ON alpha, OFF alpha and ON–OFF RGCs had varied cell responses to focused and defocused images (**F**). ON alpha, OFF alpha and ON–OFF RGCs were identified based on their response profiles to lights ON and OFF. Inhibitory (red) and excitatory (blue) currents measured in voltage-clamped ON alpha GC, OFF alpha GC and ON–OFF GC (**A–C**) holding potential −68mV and 0 mV in response to focused and equal to different dioptric powers of optical defocus ± 20D as indicated. Light stimuli of 1 s, 0.002 cycles/degree light stimuli (light intensity = 5.09 × 10^4^ Rh*/rod/sec) was projected on the outer segment of the photoreceptor layer. Defocused images had significantly different effects on EPSCs and IPSCs responses in these cells (**D**,**E**). ∗∗ *p* < 0.01; ∗ 0.01 < *p* < 0.05; n.s. *p* > 0.05.

**Figure 6 cells-09-00530-f006:**
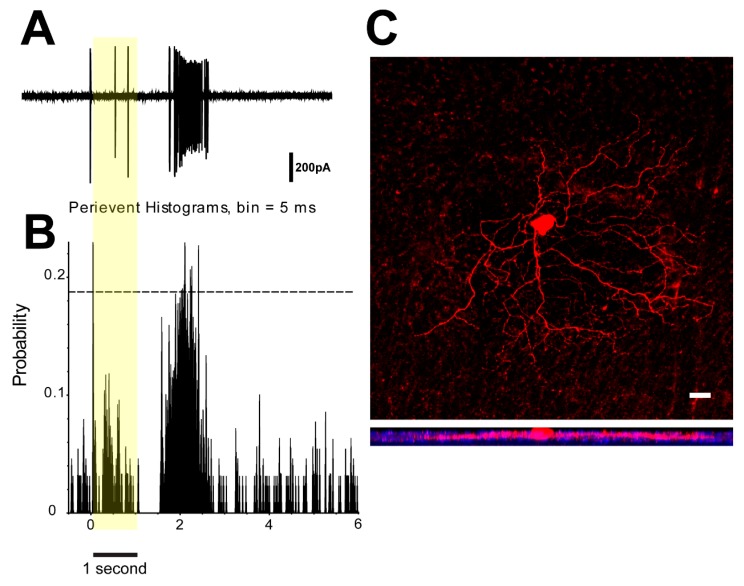
OFF-delayed RGC light response and morphology. (**A–B**) The single cell recording of OFF-delayed RGC (525-nm full-field, light intensity 1311 Rh*/rod/sec, light stimulation time 1 s, at 5 s interval) and Peristimulus time histogram (PSTH) of cell response. (**C**) The cell was visualized by Neurobiotin injection (red) and double labelled with anti-ChAT antibody (blue). Scale bar 20 µm.

**Figure 7 cells-09-00530-f007:**
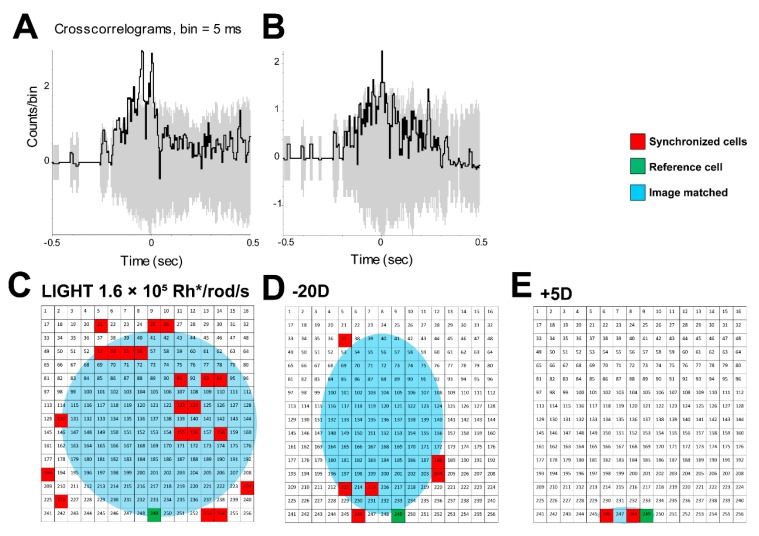
OFF-delayed RGCs/ displaced amacrine cell (dAC) synchronized firing may contribute to image edge detection. Two synchronized firing patterns (**A**,**B**) and the mapping of spatial firing pattern of focused image (**C**) and defocused image equivalent to −20D (**D**) and +5D (**E**) dioptric power were shown. The green colored box represents the reference cell, whereas the red colored box represents the synchronized cells. The highlighted blue part is the representation of edge of the image. In this experiment, only OFF-delayed RGCs/dACs were mapped. The gray part in Figure 7A,B showed shift predictor cross-correlogram profiles computed from the pairs of OFF-delayed RGCs/dACs that had no coherent firing. The image was projected on the mouse retina with diameter 1.804 mm; spatial frequency 0 cycle/degree; light intensity 1.6 × 10^5^ Rh*/rod/sec. MEA array size is 100/30 µm.

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
