# Peer review of "Defocused Images Change Multineuronal Firing Patterns in the Mouse Retina"

_cells, 2020, doi:10.3390/cells9030530_

Round 1

Reviewer 1 Report

Myopia is a public health problem. However, the underlying mechanisms of myopia are still unclear, which prevents making therapeutic treatment strategies. It is currently thought that the retina controls the development and progress of myopia.  This manuscript provides evidence that changes in retinal ganglion cell activity to defocused images could contribute to the retinal mechanisms of myopia. The data present are promising and the paper is well written. I have some minor concerns and suggestions for the authors to revise the paper.

Abstract:

Line 15: Add “(RGCs)” after retinal ganglion cells

Line 19: remove “(RGCs)”

Line 19: RGCs/amacrine cells---I feel this is confusing here and throughout the paper. I understand that your MEA recordings were from the cells in the ganglion cell layer. These cells could be ganglion cells or displaced amacrine cells. However, we know all of the ganglion cells, which consists of 49% cells in the ganglion cell layer, fire action potentials. Although 51% of the cells in the mouse retina are displaced amacrine cells, there are not many of them firing action potentials in nature. In this case, I guess the chance you record from displaced amacrine cells that fire action potentials are quite small. I would present them as ganglion cells instead of RGCs/amacrine cells throughout the paper. I would justify it either in the Methods or the Results section. In the Discussion section, I would discuss that the data could include some displaced amacrine cells, which may affect the interpretation.    

Introduction

Dopamine is part of this study. However, only a limited background for this modulator is provided in the Introduction. Readers would be appreciated if the source of ocular dopamine, regulation of dopamine release, and distribution of dopamine receptors are discussed in this section.

Line 81: remove “(RGCs)”

Line 86: change “hypothesize” to “conclude”

Results

Line 100: You mention the 5 major types are identified. However, immediately after this sentence, two other types of cells are described. These two types are present in several figures. If just look at figures, the 5 types include ON, OFF, ON-OFF, ON delayed, and OFF delayed. Clarify the classification of the cell types and make them consistent in the text and figures.  

P126: Change “ion” to “in”

P260: Change “cells” to “cell”

Figure 2L: Double check if there really is a significant difference between 0D and +10D for ON cells.

Discussion

I am a little surprised that D1R and D2R antagonists had the same effect so D1R and D2R agonists. The reason is that D2 receptors are mainly distributed in dopamine amacrine cells. The D2R agonist should suppress dopamine cell activity, blocking dopamine release onto ganglion cells. Therefore, the effect of D2R agonists on ganglion cells is assumed to the same as D1R antagonist because ganglion cells express D1Rs. Any thoughts about this?   

Reviewer 2 Report

Myopia and Hypermetropia are highly penetrant visual defects affecting large percentages of the human population.  The major issue with these conditions is that the optic apparatus of the eye (lens and cornea) does not generate a focused image at the level of the retina. This could be caused by defects in adaptive power of the lens, curvature of the cornea, or mismatch between the diameter of the eye ball and the optimal focal plane. It is believed that the retina can signal the lack of focused image to ocular and extraocular mechanisms that can in turn modify the eyeball diameter along the optic axis to accommodate the focal planes. One possibility is that Retinal Ganglion Cells (RGCs), the cells that carry the visual signal from the eye to the brain, could be involved in this corrective mechanism, or alternatively participate in the pathogenesis.  

Banerjee et al address a very interesting question: “how do defocused images affect Retinal Ganglion Cell responses?”  They do this by projecting focused and defocused images onto retinal preparations ex vivo, and recording RGC responses either through Multi Electrode Arrays (in large populations) or patch clamp recordings from individual RGCs.

The approach is innovative, and the experiments are very interesting. However, there are several manuscript delivery issues and/or methodological omissions that make the paper hard to follow, and clearly not publishable in its current version.

The authors do not clearly describe the light path through which they have presented images to the retina. The methods refer to a water immersion objective, but the dissection methods states that the sclera was not removed from the retina. Thus, given that the RGCs are facing down in MEA setup, the dark sclera would be inbetween the light stimulus and the photoreceptors, preventing light stimulation of the retina. The authors exemplify their visual stimuli as syne gratings at 0.2 Cyc/degree vs. 0.5 Cyc/degree, and show examples (Figure 2A bottom panels). The 0.5 cycles / degree should be a far finer grating compared to the 0.2.  However, in the presented image, one sees only one oval shaped area.  Do they mean 0.05 cycles/degree ? The defocusing method is not clearly explained. Here are some concerns. In my experience, an “out of focus” image has both less luminosity and different geometric features in the focal plane of the retina. This necessarily means that the number of engaged RGCs could be less (fewer are stimulated, depending on how the experiment was conducted), and also different cells on the array are excited, based on differential geometry of the “defocused stimulus”.  Thus, it is not surprising that different units and different types of responses (“ON” vs. “OFF” or “ON-OFF” responses) are observed. The analysis could be improved significantly by spike sorting all RGCs from the same preparation under all stimuli presented, and focusing the analysis on reporting the outcomes for RGCs that are followed through the various stimulation paradigms:

e.g. “Unit 25 has exhibited ON sustained responses under focused stimulus, OFF responses under +10D , ON-OFF responses under -10D, etc.”  This would actually reveal how specific RGCs have changed their response properties under different stimulation paradigms, as opposed to just reporting global shifts in numbers and properties of RGCs, which might be trivially explained by light intensity and stimulus geometry changes associated with defocusing.  Somewhat analogously, this point is made by their comparison of 0.2 and 0.5 cyc/degree experiment, where the number and distribution of units changed based on the geometry of the stimulus.

Associated with this point, the authors could mount a camera in the light path to record examples of how their defocused stimuli look, and use a photometer to understand the differences in light intensity induced by defocusing. More generally, the presentation of results has to be thoroughly reshaped to explain the goals of each experiment, what the control and what the experimental condition showed and what the major different outcomes were, section by section. This would make the manuscript much easier to follow. Although the authors have provided many experimental details in the results section, it is still hard to understand what each experiment and section convey (e.g., different RGC types are stimulated by stimuli with different spatial frequency, but the same RGCs can be recovered if the same stimulus is re-applied (0.2 vs. 05 cyc/d experiment – Figure 2 A- F). What is the purpose of the pharmacological manipulations of DOPAmine receptors ? Gating of spatial resolution via control of Gap junctions ? English has to be revised as some serious grammar mistakes are still present, as well as misuse of some words (example line 94 = 525 nm “full filed” should be “full field”)

Reviewer 3 Report

Manuscript summary & General comments:

In this study, the authors demonstrated that defocused images induce the alterations in the multi-neuronal firing patterns and whole-cell conductance in the mouse retina. Also, they addressed that the multi-neuronal patterns are closely related to dopamine receptors and off-delayed ganglion cells (GCs) may contribute to edge detection. Generally, the study was well-designed, and electrophysiological methods were appropriately applied to reveal the cause of myopia at the retina level. However, it appears not to be a brand new study. Rather it appears to be an attempt to further corroborate their previous beautiful study (Pan, 2018) titled "Defocused image changes the signaling of GC in the mouse retina” by MEA recording and analysis. Finally, there are a few limitations and questions of this study, considering its purpose of suppressing and preventing myopia progression according to the development of the retina.

Specific comments:

Discussion, Page 7, 337~340

The authors noted that edge detection can be associated with synchronous firing of off-delayed GCs, suggesting that it could be a mouse specific property due to the opposite result in chicken retina (Diedrich & Schaeffel, 2009). But if such MEA recording is uniquely detected from off-delayed GCs that are highly correlated with myopia, shouldn't it be applied to animal specificity? If the difference is due to species specificity, is it correct to define that the result of detection of the myopic retina (GC response) using MEA recording varies from animal to animal? More importantly, the author's research aims to suggest ways to prevent myopia in humans. How decisive can a mice’s firing pattern be in revealing the cause of myopia in humans? While suppressing the rod pathway in mice (specialized for night life), the use of (very limited) cone response is thought to be too artificial to apply to humans (day vision is more important).

2. Authors argued an importance of early detection and prevention of development of myopia. Have you ever found age-dependent symptom such as different firing patterns observed for different aged mice in response to the same defocused image?

Method, page 9. line 406

Many researchers commonly use amphotericine or gramicidin in perforated-patches, while authors used β-aescin. Do you have any specific reason why you use β-aescin to add to the cells? Did you expect any benefit by use of β-aescin? Please explain how to use it.

3 page line 126,

“but not ion the same position” should be corrected.

5 page line 199,

“were firedat…” should be corrected to “were fired at..”

Round 2

Reviewer 1 Report

The authors have done an excellent job for this revision and I do not have further comments.

Author Response

Dear respected reviewer,

I sincerely thank for your constructive criticisms and valuable comments.

With my best wishes

Feng Pan

Reviewer 2 Report

  • Light Path and retina preparation The Authors have nicely explained the retina preparations in their response to reviewers but have not done so in the text of the manuscript. A more complete description of retina prep and light paths should be provided.  The readers will not have the reviewer responses available to understand the experiment.

  • Calculation of Stimulus Spatial Frequency. The authors provide the sample code for their images and example images. I am not familiar with the way PsychoPy encodes its stimuli, however, what is obvious is that there is something wrong with the “5 C/D =50/100” condition.  By definition, the higher the C/D ratio, the greater the number of stripes/ degree. This is obvious if you compare the 0.2 and 0.05 condition, where more stripes are visible in the 0.2 condition.  However the 0.5 C/D looks like a homogeneous gray screen.  Is it possible that this setting is beyond the resolution of the OLED screen, and hence it is giving you just the average gray?  If so, the results should be presented and interpreted accordingly in the text. Instead of talking about 0.5 C/D, simply talk about a uniform gray screen. This then has a significant influence on the interpretation of the data, as the 0.5 C/D image is essentially devoid of any edges, so there is no spatial information conveyed in it, other than uniform luminance.

  • Effects of defocusing on light stimulus

Again, the authors have done a much better job explaining their experiment in the response to review letter compared to the main body of the manuscript. It would be useful to include the example pictures shown in the response to reviewer letter in the manuscript, perhaps by generating an additional figure under the methods section, to show the effects of “out of focus” images. Alternatively, the defocused images could be shown as insets in the relevant figures.

  • Effect of defocusing on RGC response properties. Since the authors agree with the interpretation that the local stimulus conditions change under defocusing in comparison with focused images, they should address this in the discussion in an explicit way. They now show that only a minority of RGCs can be detected under all stimulus conditions. That has a trivial explanation, namely that the different stimulus intensity and geometry at the same location results in the stimulation of different RGC populations, and hence the retinal response is distinct. The really interesting cells are the ones that respond across all stimuli, including focused and defocused. Are their receptive fields integrating over larger areas, hence they do not care about the loss of spatial detail?

Reviewer 3 Report

The authors adequately answered my questions. I accept all the answers.

Author Response

(The authors gave the same response as above.)
